# Comparing Internal Flow in Freezing and Evaporating Water Droplets Using PIV

**Linn Karlsson \*,†** , **Anna-Lena Ljung †** and **T. Staffan Lundström †**

Division of Fluid and Experimental Mechanics, Luleå University of Technology, SE-97187 Luleå, Sweden;
anna-lena.ljung@ltu.se (A.-L.L.); staffan.lundstrom@ltu.se (T.S.L.)
* Correspondence: linn.karlsson@ltu.se
† These authors contributed equally to this work.

**Abstract:** The study of evaporation and freezing of droplets is important in, e.g., spray cooling, surface coating, ink-jet printing, and when dealing with icing on wind turbines, airplane wings, and roads. Due to the complex nature of the flow within droplets, a wide range of temperatures, from freezing temperatures to heating temperatures, have to be taken into account in order to increase the understanding of the flow behavior. This study aimed to reveal if natural convection and/or Marangoni convection influence the flow in freezing and evaporating droplets. Droplets were released on cold and warm surfaces using similar experimental techniques and setups, and the internal flow within freezing and evaporating water droplets were then investigated and compared to one another using Particle Image Velocimetry. It was shown that, for both freezing and evaporating droplets, a shift in flow direction occurs early in the processes. For the freezing droplets, this effect could be traced to the Marangoni convection, but this could not be concluded for the evaporating droplets. For both evaporating and freezing droplets, after the shift in flow direction, natural convection dominates the flow. In the end of the freezing process, conduction seems to be the only contributing factor for the flow.

**Keywords:** evaporation; freezing; water droplet; internal flow; Marangoni flow

---

## 1. Introduction

Evaporation and freezing of droplets are two interesting areas with many applications. Evaporation of droplets is important in, e.g., spray cooling [1], surface coating [2], ink-jet printing [3], and droplet based biosensing [4]. For freezing droplets, the applications are found mainly when dealing with icing on wind turbines [5], airplane wings [6], and roads [7]. Since icing is a big problem for these areas, effective icing prevention methods have been developed, but a lot is still unknown about the ice itself. One way in improving these existing icing prevention systems is to go back a few steps and start with a single freezing water droplet. Due to the complex nature of the flow within droplets, a wide range of temperatures, from freezing temperatures to heating temperatures, have to considered to increase the understanding of the flow behavior. This is the motivation for performing this comparative study between freezing and evaporating droplets.

To the author's best knowledge, not many studies have been performed studying the internal flow within the droplets for freezing of water droplets. Kavanami et al. [8] used a numerical model considering both surface tension and the density maximum at 277 K. The results were also validated with experiments. They found that both natural and Marangoni convection are important mechanisms for the internal flow. Karlsson et al. [9] modeled the inner flow when internal natural convection was included and excluded (i.e., only conduction from the surface) in the model and the result was validated with existing experimental data. Karlsson et al. [10] performed experiments by releasing

droplets on a cold surface and the inner flow was captured by using Particle Image Velocimetry (PIV). The flow field in the center of the droplet was visualized and the magnitude and direction of the velocities were derived.

The evaporation of sessile droplets has been studied thoroughly both experimentally and numerically [11–15]. A similar setup as in the experiments performed in this study was done by Erkan and Okamoto [16] and Erkan [17], where liquid droplets were released with different impact velocities onto a dry (and later, also heated) sapphire plate. The flow inside the droplet was investigated using time-resolved PIV, and a laser created a laser sheet illuminating a plane parallel to the surface. The radial velocity distribution during the early phase of spreading inside the droplets was compared to an analytical as well as a numerical model and both were proven to show similar results. However, none of the models could reproduce all of the experimental results in detail.

An interesting study of the internal flow within droplets impacting on a surface was performed by Kumar et al. [18] using the same ray-tracing algorithm as in this work proposed by Kang et al. [19] to account for the image distortion at the droplet boundary. The vortices caused by the impact to the surface were traced and the vortex strength was computed. The internal flow was also studied by Thokchom et al. [20,21] using PIV in externally heated sessile droplets confined in a narrow gap between two glass plates. The benefit of using a "Hele–Shaw" droplet is in the optics since no image correction has to be done and the surface flow is seen. The experiments yielded that an alteration in the free surface temperature changes the fluid flow profile inside the droplet and consequently also the deposition pattern of solute particles on the substrate. The heat sources were an IR light and a heating element that was moved along the droplet surface to create different effects. The measured velocity profiles were in qualitatively agreement with numerical simulations.

In a recent study by Zhao et al. [22], the full velocity field in evaporating sessile droplets was captured. Instead of correcting the raw PIV data as done in this work, the Scheimpflug principle was applied. This is a mapping method used to eliminate the perspective effects and therefore the flow near the droplet surface can be studied. This method is very useful when studying evaporating droplets, but when working with freezing droplets this method will not work properly because of the positioning of the camera. Since the camera is placed underneath the droplet, the frost layer and the created ice during freezing will make it impossible for the camera to retrieve good images of the freezing process. This is one of the reasons the method by Kang et al. [19] was used in the freezing experiments in the current study. For easier comparison, the same method was also applied in the evaporation experiments.

In this work, the effects of Marangoni convection on evaporating and freezing droplets were studied and the results from the two different processes were compared to each other. The effects of Marangoni convection on evaporating droplets has been studied before. To exemplify, Hu and Larson [23] concluded from numerical simulations that a thermally driven Marangoni flow should be visible in an evaporating water droplet, but this is not as easy to visualize in experiments. One possible reason for this can be surfactant contaminants [23,24]. Xu and Luo [25] showed that, even though this is partially true, the surface tension and the surface temperature change nonmonotonously along the liquid surface due to a stagnation point at the droplet surface. Kita et al. [26] induced a Marangoni flow in pure water drops by creating a temperature gradient on the drop using infrared thermography. They found that, to initiate a Marangoni flow as two similar vortices, a temperature gradient along the liquid–air interface of about 2.5 K (2.5 °C) is necessary and to maintain them the temperature difference has to be about 1.5 K (1.5 °C). The main consensus is that Marangoni convection is small for water droplets [24,27], but can be induced [26]. This motivates also a comparison between heated evaporating droplets and freezing droplets since it has been shown that the Marangoni convection plays an important role for the flow in the beginning of the freezing process, but after a while natural convection takes over [10]. This is not as clear when studying evaporation of droplets, and this paper discusses the different effects in the droplets.

Hence, in this study, the aim was to investigate and compare the inner flow when droplets are released on a cold and warm surface and to reveal if natural convection and/or Marangoni convection have a noticeable influence on the flow within the droplets.

## 2. Method

To facilitate the presentation of the results, a schematic image of the direction of Marangoni flow for a constant gradient is shown in Figure 1. The Marangoni flow when $T_1 > T_2$ is seen to the left in the figure, i.e., the flow within an evaporating droplet, and the Marangoni flow when $T_1 < T_2$ is seen to the right, i.e., the flow within a freezing water droplet.

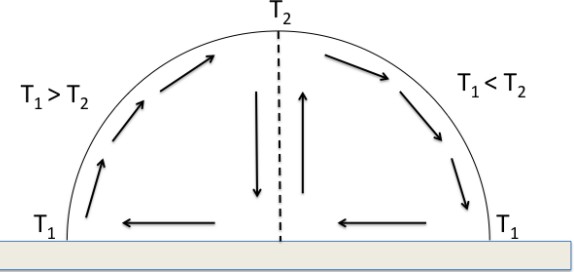

**Figure 1.** Direction of Marangoni driven flow for a constant gradient: (**Left**) $T_1 > T_2$, evaporation; and (**Right**) $T_1 < T_2$, freezing.

A similar setup and experimental procedure were used for the freezing and evaporating droplets. When these differ from one another, it is clearly stated in the text.

### 2.1. Experimental Setup

Droplets were gently deposited on a sapphire plate (aluminum oxide, $Al_2O_3$) using a pipette. The sapphire plate measured 0.0508 m (2″) in diameter, 0.005 m thick, and was chosen due to its high thermal conductivity in combination with its transparency. The pipette was a ThermoFisher Finnpipette F1 1–10 μL, and it was kept in place by an optical rail to create repeatable deposition of the droplets in each experiment. The sapphire plate was resting on an aluminum holder that was heated using a Peltier element. The Peltier element was placed with its warm side up during the heating experiments and its cold side up during the freezing experiments. It was submerged in a box with either warm or cold circulating water depending on which experiment was performed (closed system, connected to a tank water held at $T = 295.85$ K $\pm$ 0.98 K and $T = 277.05$ K $\pm$ 1.3 K, respectively). The laser light was guided underneath the droplet using a prism that was put inside a central hole in the aluminum holder and a tunnel inside the holder allowed the light to pass through to the droplet. To protect the setup from disturbances in the room, a PMMA (plexiglass) chamber was put around the sapphire plate. Four holes was made in the chamber; one on top for the pipette to be able to release the droplets, one on the right side to allow the laser light to pass through to the prism, one on the left side to enable the camera to record the experiments and one on the back where the hygrometer was mounted. The sapphire plate was cleaned using Propanol ($C_3H_7OH$) and deionized (DI) water using lens paper before each experiment. A schematic diagram of the experimental setup can be seen in Figure 2. In both heating and freezing experiments, the liquid used was DI water kept at room temperature, $T_{water} = 294.25$ K $\pm$ 1.4 K. The seeding particles in the freezing experiments were chosen due to its good qualities when working with water. However, these particles did not work well in the heating experiments since the droplets collapsed as they reached the surface after the release from the pipette. Therefore, new particles were chosen for the heating experiments. Table 1 presents details of the seeding particles for each setup. The Stokes number, *Stk*, was calculated for a "worst-case scenario" for both setups to determine if the particles was suitable for these experiments (see Table 1). Since both have a *Stk* « 1, the conclusion is that the particles follow the flow well. Based on the guidelines,

the amount of particles in the water was continuously evaluated to reach the recommended particles per interrogation area. The resulting concentration of particles for both setups can be found in Table 1.

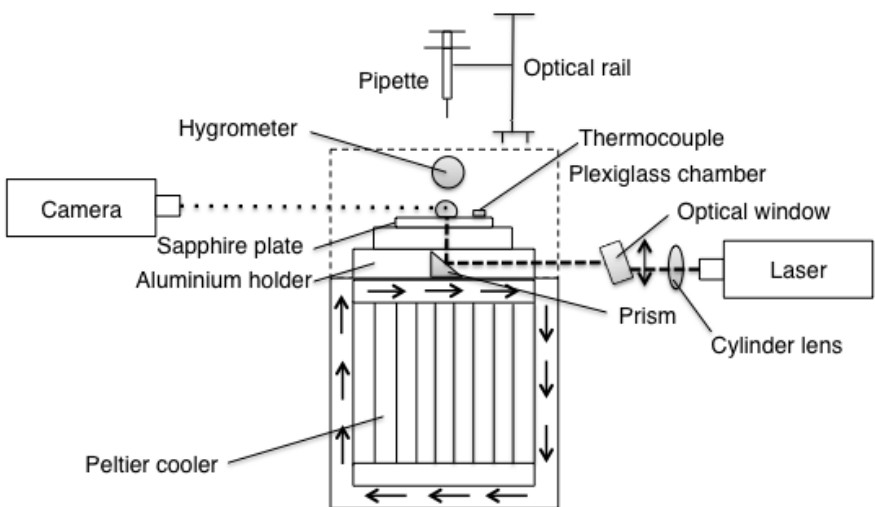

**Figure 2.** Schematic diagram of the experimental setup.

**Table 1.** Data of seeding particles.

| Case | Particles | Diameter, $d$ (m) | Stk | Conc. of Particles |
|---|---|---|---|---|
| Heating | Latex (Magsphere Inc., Pasadena, CA, USA PS Red Fluorescent, 1055 kg/m³) | $5.1 \times 10^{-6}$ | $<5 \times 10^{-4}$ | $9.9 \times 10^{-7}$ m³ DI water and $0.1 \times 10^{-7}$ m³ seeding particles |
| Freezing | Rhodamine B (microParticles GmbH PS-FluoRed, 1050 kg/m³) | $3.16 \times 10^{-6}$ | $<5 \times 10^{-7}$ | $9.8 \times 10^{-7}$ m³ DI water and $0.2 \cdot 10^{-7}$ m³ seeding particles |

The temperature of the air and the humidity inside the chamber were monitored using a hygrometer and the temperature of the sapphire plate was measured using a thermocouple of K-type. The laser was a continuous 50 mW 532 nm Nd:YAG (Altechna Co Ltd., Vilnius, Lithuania) connected to a half wave-plate, a polarizing beam splitter (cube) and a beam dump. These components were used to adjust the amount of light transmitted through the droplet. A cylinder lens assembly from Dantec Dynamics created and focused the light sheet to a thickness of <0.0004 m. A 0.012-m-thick window placed on a rotation table was used to fine tune the position of the sheet to the center of the droplet (up and down to be able to adjust the light sheet to move sideways in the droplet) (see Figure 2). The motivation for guiding the laser light from underneath the droplet was the advantage of the plane surface reducing the light scatter. This arrangement also allows a good view of the symmetry line of the droplet. For the freezing setup, a CMOS camera (IDS µEye) with a spatial resolution of 1280 × 1024 pixels and pixel size $5.3 \cdot 10^{-6} \times 5.3 \cdot 10^{-6}$ m² together with a Navitar long distance microscope captured images of the particles. For the heating setup a similar CMOS camera (IDS UI-3140CP-M-GL Rev.2) was used with the same spatial resolution and microscope mounted, but with pixel size $4.8 \cdot 10^{-6} \times 4.8 \cdot 10^{-6}$ m². Details about the specific sampling rates can be found in Table 2. The height from which the droplets were released was measured to be 0.0021 m and the velocity as the droplet hit the surface was calculated to be 0.052 m/s for the heating experiments and 0.0039 m and 0.077 m/s, respectively, for the freezing experiments.

**Table 2.** Experimental setup conditions.

| Case | Plate Temperature, $T_{plate}$ | RH in Chamber | Temperature in Chamber, $T_{chamber}$ | Sampling Rate | Recording Times |
|---|---|---|---|---|---|
| Heating | 313.15 K ± 0.22 K, 323.15 K ± 0.11 K, 333.15 K ± 0.05 K | 47.2% ± 1.2% | 298.55 K ± 0.77 K | 50–54 Hz | 60 s |
| Freezing | 265.07 K ± 0.12 K, 261.11 K ± 0.06 K | 50.4% ± 4.5% | 289.85 K ± 1.7 K | 50, 54 Hz | Dependent on the freezing times of the droplets |

*2.2. Experimental Procedures*

The following statements can briefly summarize the experimental procedures:

**Heating**

1. The heating of the surface started.
2. At $T_{plate}$ = 313.15, 323.15 or 333.15 K (visually determined from the computer screen):

    - The pipette was filled with the DI water and seeding particle suspension.
    - The camera and the laser were switched on.
    - The laser sheet was fine tuned to the center of the droplet (while the droplet was still hanging from the pipette).
    - Finally, the droplet was released.

3. The camera and laser light were turned off after about 60 s after the droplet has hit the surface.
4. The surface was cleaned and a new experiment could begin.

**Freezing**

1. The cooling of the surface started.
2. At $T_{plate}$ = 261.15 K or 265.15 (visually determined from the computer screen), the pressurized air was switched on and turned off again when RH was around 50%. This took about 60 s.
3. The pipette was filled with the DI water and seeding particle suspension.
4. The camera and the laser were switched on. The position of the light sheet was fined tuned to the center of the droplet (while the droplet was still hanging from the pipette).
5. The droplet was released when $T_{plate}$ reached 265.15 K (or 261.15 K) again (visually determined from the computer screen), which occurred approximately 30 s from when the pressurized air was switched off.
6. The cooling was turned off when the droplet was completely frozen.
7. The surface was cleaned and dried when $T_{plate}$ > 273.15 K and at $T_{plate}$ = 277.15 K a new experiment could begin.

In the freezing experiments, it was found that a roughness on the surface was necessary to initialize the freezing process without delay. Due to the smoothness of the sapphire plate, this roughness was added by producing frost on the surface. This layer was generated by letting pressurized air pass through a container filled with water and this humidified air was then guided into the closed chamber surrounding the experimental setup. The temperature of the surface and air and the relative humidity inside the chamber were monitored to produce a layer of frost as similar as possible during each freezing (see Table 2 for details about the conditions for the frost). A longer discussion of how this frost layer can be found in the work of Karlsson et al. [10].

The result of the evaporation experiments is based on six evaporating droplets with a similar geometry, i.e., droplets with approximately same heights and radii, when $T_{plate}$ = 313.15, 323.15, and 333.15 K. These are presented in Table 3. Two freezing droplets (when $T_{plate}$ = 261.15 and 265.15 K) are also included in this table.

**Table 3.** Values of the geometries of the evaporating and freezing droplets at $t = 1$ s.

| Case ($T_{plate}$-Temperature) | Droplet Height, $h$ (m) | Droplet Radius, $r$ (m) | Contact Area at Surface, $A$ (m$^2$) |
|---|---|---|---|
| **Evaporation** | | | |
| 313.15 K: Case 1 | 0.00137 | 0.00188 | $11.1 \times 10^{-6}$ |
| 313.15 K: Case 2 | 0.00143 | 0.00186 | $10.9 \times 10^{-6}$ |
| 323.15 K: Case 1 | 0.00145 | 0.00186 | $10.9 \times 10^{-6}$ |
| 323.15 K: Case 2 | 0.00147 | 0.00184 | $10.6 \times 10^{-6}$ |
| 333.15 K: Case 1 | 0.00148 | 0.00192 | $11.6 \times 10^{-6}$ |
| 333.15 K: Case 2 | 0.00145 | 0.00185 | $10.7 \times 10^{-6}$ |
| **Freezing** | | | |
| 261.15 K | 0.00178 | 0.00156 | $7.65 \times 10^{-6}$ |
| 265.15 K | 0.00142 | 0.00171 | $9.15 \times 10^{-6}$ |

*2.3. Uncertainty Analysis*

The uncertainties introduced during the measurements can be divided into two categories: systematic (bias) and random errors. The systematic errors usually arise from the measuring equipment and the random errors are usually due to unknown or unpredictable changes in the experiments [28].

Similar to the experiments performed by Karlsson et al. [10], the main sources of the systematic errors can be found in the pipette technique, i.e., the mixing of seeding particles in the water, the reading of the measurements instruments, and the positioning of the camera. Since these errors are introduced in each experiment, they may be difficult to detect and therefore have a large impact on the result. However, careful planning and execution of experiments can minimize these errors. Normally, the correlation error is 0.1 pixel [29].

The random errors can mainly be found in the release of the droplet, resulting in droplets with different geometries and different initial internal flows ($t < 5$ s). To get an estimate of the random errors, a repeatability test can be performed (see [28]). Ten experiments for $T_{plate} = 333.15$ K were considered (see Table 4 for details about the droplets), where the magnitude of the corrected velocities in the y-direction along the symmetry line between bottom and apex of the droplet was studied. Since the droplets varied in height, the interesting points is what happened at just above the heated surface, i.e., the lowest value of the corrected dataset and at the top of the corrected data in each case. In addition, the points at 25%, 50%, and 75% above the heated surface were determined. To get an estimate of the variations in the velocities along the symmetry line in the beginning of evaporation and as the flow settled in the droplet, the times studied were $t = 5$ and 15 s. The specific time $t = 5$ s was chosen because the shift in flow direction (see Section 3.2.1) had occurred for all 10 cases in this investigation; before this time the flow would be more difficult to evaluate. In Table 5, the precision errors with a 95% confidence interval for the five points chosen are shown and it can be seen that the errors are below, or mostly well below 5.5%, suggesting that the random errors are relatively small regarding the velocities on the symmetry line, especially for when $t = 15$ s. This can also be seen in Figure 3, where the mean velocity in the five points is shown together with the precision error presented using error bars. This means that the velocities in the droplet are in fact comparable in each case despite their differences in geometry, and the six selected droplets in Table 3 can be used in the further study.

**Table 4.** Values of the droplets geometry in the repeatability study at $t = 5$ and 15 s when $T_{plate} = 333.15$ K. Note that the radius of the droplets and contact areas at the heated surface are the same for both times.

| Case | Droplet Radius, $r$ (m) | Contact Area at Surface, $A$ (m²) | Droplet Height, $h$ at $t = 5$ s (m) | Droplet Height, $h$ at $t = 15$ s (m) |
|------|---------|-----------------|----------|----------|
| 1 | 0.00180 | $10.1 \times 10^{-6}$ | 0.00143 | 0.00139 |
| 2 | 0.00186 | $10.9 \times 10^{-6}$ | 0.00137 | 0.00133 |
| 3 | 0.00191 | $11.5 \times 10^{-6}$ | 0.00149 | 0.00144 |
| 4 | 0.00178 | $9.98 \times 10^{-6}$ | 0.00157 | 0.00152 |
| 5 | 0.00189 | $11.2 \times 10^{-6}$ | 0.00143 | 0.00138 |
| 6 | 0.00186 | $10.8 \times 10^{-6}$ | 0.00145 | 0.00142 |
| 7 | 0.00187 | $11.0 \times 10^{-6}$ | 0.00144 | 0.00140 |
| 8 | 0.00177 | $9.80 \times 10^{-6}$ | 0.00145 | 0.00142 |
| 9 | 0.00192 | $11.6 \times 10^{-6}$ | 0.00147 | 0.00142 |
| 10 | 0.00184 | $10.6 \times 10^{-6}$ | 0.00148 | 0.00143 |

**Table 5.** The precision error in five points along the symmetry line for $t = 5$ and 15 s.

| Position | $t = 5$ s | $t = 15$ s |
|----------|-----------|------------|
| At heated surface | 1.00% | 0.81% |
| 25% | 0.25% | 1.78% |
| 50% | 5.37% | 0.42% |
| 75% | 2.30% | 0.22% |
| Top | 2.68% | 0.02% |

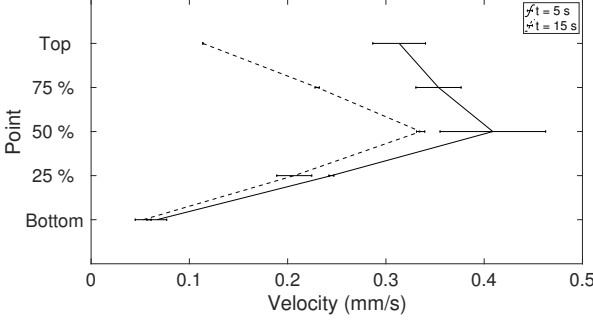

**Figure 3.** The mean velocity in the five points used in the repeatability study at $t = 5$ and 15 s for 10 experiments when $T_{plate} = 333.15$ K. The precision error with a 95% confidence interval in each point is shown with error bars. Point 1 is located at the heated surface and Point 5 is located at the top of the corrected data. Points 2–4 are found in between Points 1 and 5.

## 3. Results and Discussion

### 3.1. Impact of Release

When a droplet of equal temperature as the sapphire plate, held at $T = 293.15$ K, is released and reaches the surface, all movement stops completely within less than a second. This could suggest that the movement inside the droplet after $t = 1$ s does not have to do with the release of the droplet. However, since the viscosity of the water decrease with an increase in temperature (drops about 50% from 293.15 to 333.15 K), any movement created by the release of the droplet should be more visible at higher temperatures. This indicates that depending on how the droplet impact the surface, different types of flow will be seen inside the droplets and there will be a seemingly random motion in each droplet directly after impact. This can be compared to the freezing droplet where the viscosity in the water is high as it reaches the surface and will only get larger as the temperature inside the droplet decreases. This means that the movement inside of the freezing droplet is not induced by the release

of the droplet, but for the heated droplet the release may have a larger impact on the flow inside the droplet.

*3.2. Evaporation*

After the droplet is released on the surface the flow is fluctuating in velocity and for at least the next 5 s; see Figure 4 where the magnitude of the mean velocity along the the symmetry line for $T_{plate}$ = 313.15, 323.15 and 333.15 K when $t$ = 1–50 s is plotted. The size of the fluctuations and how long these occur, increase with increasing temperature, i.e., when $T_{plate}$ = 313.15 K, the velocity settles more quickly than for when $T_{plate}$ = 333.15 K. After this initial time period, i.e., when $t$ = 5–20 s, the flow settles and the velocity decreases for the higher temperatures and is fairly constant for the lowest temperature. A form of "steady state" is reached, since the conditions in and around the droplet are not changing significantly. The only varying parameter is the drying of the droplet, which reduces the size and thereby also the velocities inside the droplet. This section is divided in two parts: evaporation until "steady state" and evaporation after "steady state".

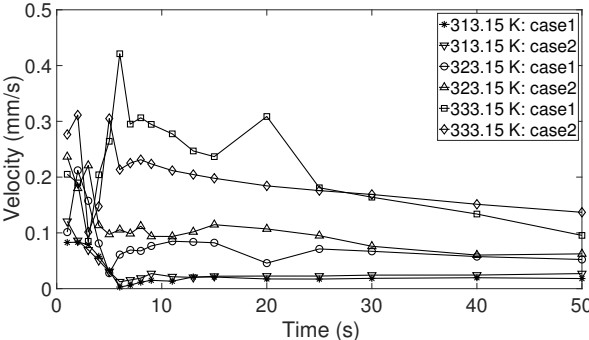

**Figure 4.** The magnitude of the mean velocity along the symmetry line for $T_{plate}$ = 313.15, 323.15, and 333.15 K when $t$ = 1–50 s.

3.2.1. Evaporation until "Steady State"

For all cases studied, during the first second or the first few seconds for some cases, the flow is moving down the symmetry line of the droplet and up along the curved surface. This is exemplified in Figure 5a for $T_{plate}$ = 333.15 K (Case 1). Note that the velocity vectors are normalized with the magnitude of the velocity of each vector. Within 15–20 s, the flow has settled and the flow is moving in the opposite direction up along the symmetry line. For $T_{plate}$ = 333.15 K (Case 1), this takes place already within 3 s. After this, the flow is moving upwards along the symmetry line and down along the curved surface. The initial direction of the flow may indicate that Marangoni convection can actually be seen in heated water droplets directly after the impact on the surface. After this initial time period, natural convection takes over changing the direction of the flow. This is interesting since the effects of Marangoni convection has also been shown to be the dominant for freezing droplets, up to about 15% of the total freezing process [10]. However, even though the direction corresponds to Marangoni flow, it is difficult to conclude that this is the cause. Another explanation for this flow pattern is due to the release of the droplet. The force created as the droplet impact the heated surface will be directed upwards in the direction away from this surface forcing the water upwards along the curved surface in the droplet. Since the viscosity is decreasing with increasing temperature, the water moves more easily when a force is applied. Finally, a third possibility for this type of initial flow pattern might be due to the temperature differences, $\Delta T$, caused by the heating of the surface. When $\Delta T = 0$ (at room-temperature), no flow is seen inside the droplet and this could indicate that the droplets initial change in form do not induce a flow. However, when $\Delta T \neq 0$, this initial change in form might actually affect the flow and this could cause the flow moving in this initial direction, as exemplified in Figure 5. All three causes are plausible and there might also be an interaction among all three contributing to the flow.

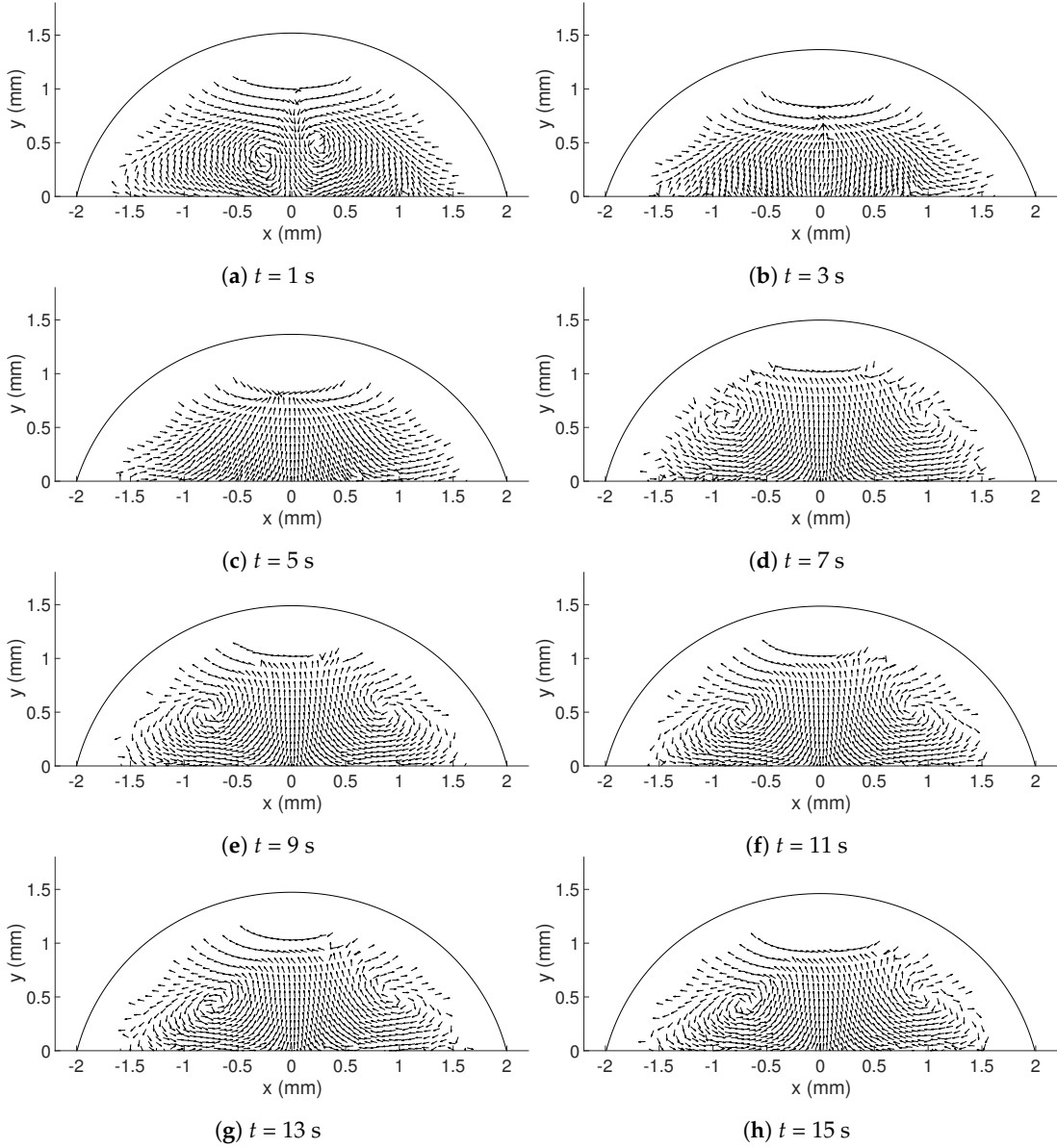

**Figure 5.** Internal flow patterns with normalized velocity vectors for the corrected data when $T_{plate}$ = 333.15 K (Case 1) at $t$ = 1–15 s.

The direction of the flow is altered within seconds after the release from the pipette. This can be seen in Table 6, where the time for when the flow shifts in direction can be found for all cases. As the temperature increases, the time for when the flow is shifting is decreasing. Since the viscosity decreases with higher temperatures, the impact from the release might be seen for a longer time in the flow or it could also be that the effects from Marangoni has larger impact on the flow when the temperature increases. It should be noted that the flow seen in Figure 5 is similar, but not identical, in each of the six cases studied. Even though the goal for the release of the droplet on the surface is to be identical in each experiment, there are still differences, and therefore different types of flow emerges inside the droplet. Vortices are created at different locations in each droplet, but, as the flow settles (at $t$ = 5–20 s), two vortices are seen or are discernible at either side of the symmetry line in the droplet (see Figure 5). The flow direction, as discussed above, is however the same in all droplets. When the turn in flow direction occur the velocities drops considerably, as seen in Figure 6 at $t$ = 3 s, where the magnitude of the velocity is exemplified for $T_{plate}$ = 333.15 K (Case 1). After the shift, the velocities recover and reach a similar level as before. Towards the end of the measuring period ($t$ = 13 s), the flow slows down. When the temperature of the heated surface is higher, the velocities are also higher inside

the droplet; see Figure 7, where the spread (standard deviation) in velocity along the symmetry line in the droplet at $t = 1$–15 s can be seen for all temperatures. Here, the mean velocity is seen as a solid and a dotted line representing the two cases for each temperature. As the temperature of the heated surface increases, the spread in velocity becomes larger in magnitude. This is due to both the release of the droplet and the large temperature differences in the droplet compared to the heated surface and is therefore expected. Some variations in velocity between the two cases compared at each temperature can be seen, but the magnitude of velocity is similar, suggesting that they are in fact comparable and repeatable as long as the droplet geometries are similar.

**Table 6.** The point in time when the flow shifts during evaporation.

| Case | Time (s) |
| --- | --- |
| 313.15 K: Case 1 | 6 |
| 313.15 K: Case 2 | 6 |
| 323.15 K: Case 1 | 4 |
| 323.15 K: Case 2 | 4 |
| 333.15 K: Case 1 | 3 |
| 333.15 K: Case 2 | 2 |

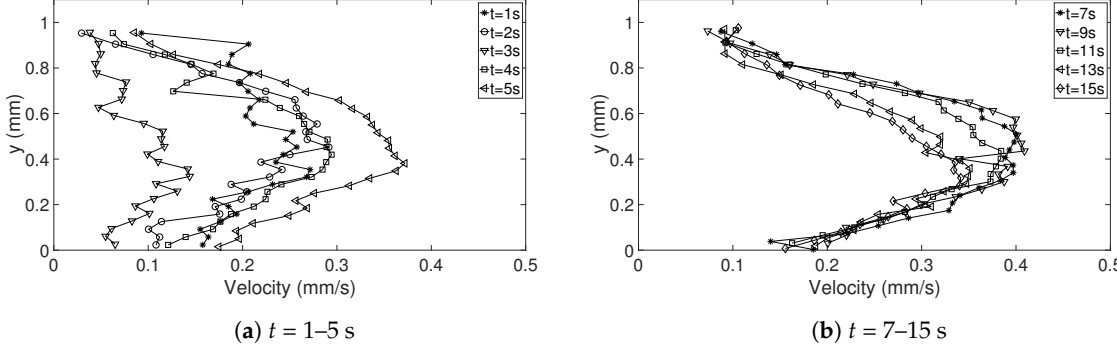

(**a**) $t = 1$–5 s　　　　　　　　　　　　　　　(**b**) $t = 7$–15 s

**Figure 6.** The magnitude of the velocity along the symmetry line for $T_{plate} = 333.15$ K (Case 1) when $t = 1$–5 s and $t = 7$–15 s.

### 3.2.2. Evaporation after "Steady state"

After the initial time period, the flow inside the evaporating droplets begins to settle. The velocity starts to decrease around $t = 15$–20 s and the decrease in mean velocity is more rapid when $T_{plate} = 333.15$ K (see Figure 4). To exemplify, if studying the velocity vectors (normalized with the magnitude of the velocity of each vector) for $T_{plate} = 333.15$ K (Case 1) in Figure 8 for $t = 20$–50 s, it can be seen that the flow is approximately similar during this time. The vortices move somewhat between the time frames, probably due to a decrease in velocity and contact angle, but a type of "steady-state" has occurred. This "steady-state" is also reflected in Figure 9, where the magnitude of the velocity along the symmetry line for the same case and corresponding times is seen and here the velocities are approximately similar with a slight decrease with time. In Figure 4, this decrease is seen for all $T_{plate}$ temperatures and cases. Scrutinizing the full time period, i.e., $t = 1$–50 s, yields that the decrease in velocity is more rapid in the beginning of the evaporation compared to the end. When the "steady-state" period is reached, the temperature differences between the water and the heated surface have evened out and the heat exchange with the surroundings is constant. In addition, the velocity differences increases with the increase in surface temperature of the heating surface if the full time period, $t = 1$–50 s is considered. In Figure 10, the spread (standard deviation) in velocity along the symmetry line in the droplet at $t = 20$–50 s can be seen for all temperatures and here the mean velocity is seen as a solid and a dotted line representing the two cases for each temperature. Note that the maximum spread in velocity is still larger (if comparing to Figure 7) when the temperature of the heated surface is higher compared to a lower temperature.

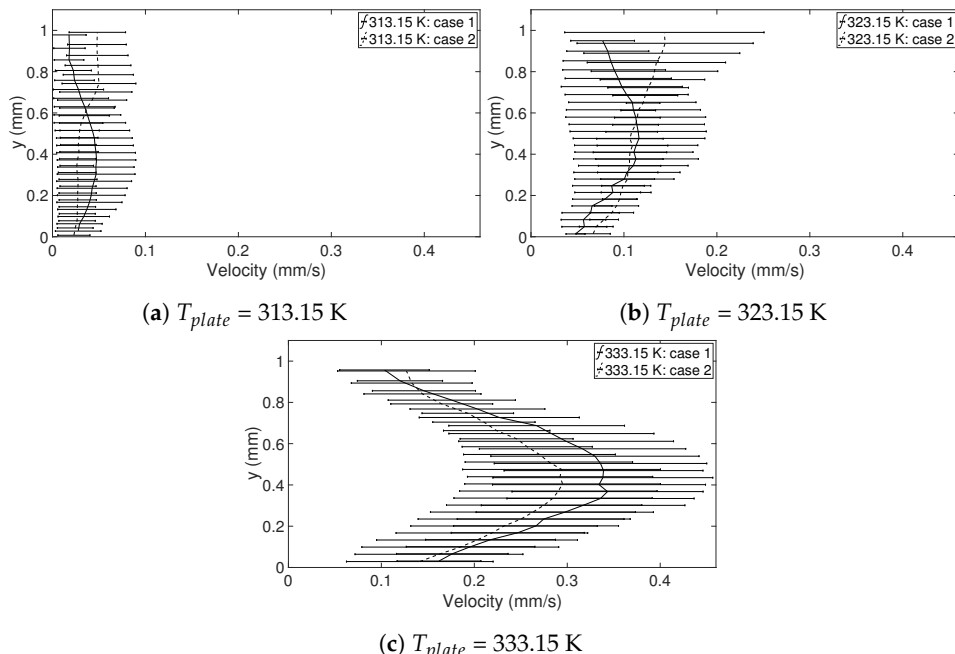

**Figure 7.** The spread (standard deviation) in velocity along the symmetry line during evaporation for $T_{plate}$ = 313.15, 323.15, and 333.15 K (all cases) when $t$ = 1–15 s shown using error bars. The solid line is the mean velocity of all times.

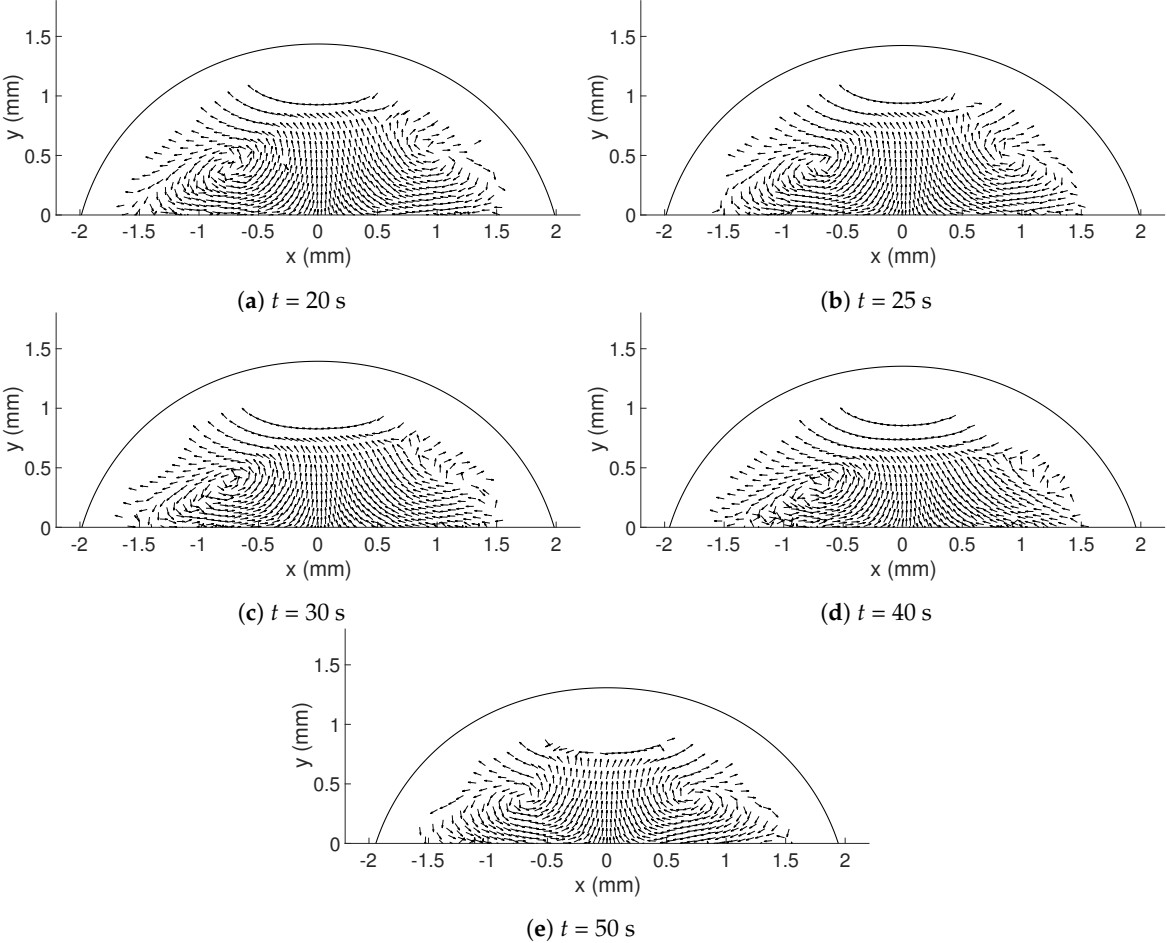

**Figure 8.** Internal flow patterns with normalized velocity vectors for the corrected data when $T_{plate}$ = 333.15 K (Case 1) at $t$ = 20–50 s.

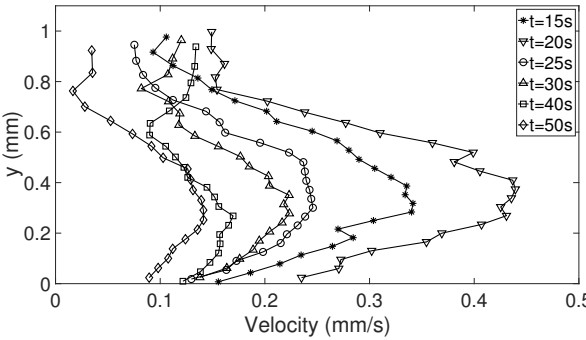

**Figure 9.** The magnitude of the velocity along the symmetry line for $T_{plate}$ = 333.15 K (Case 1) when $t$ = 20–50 s.

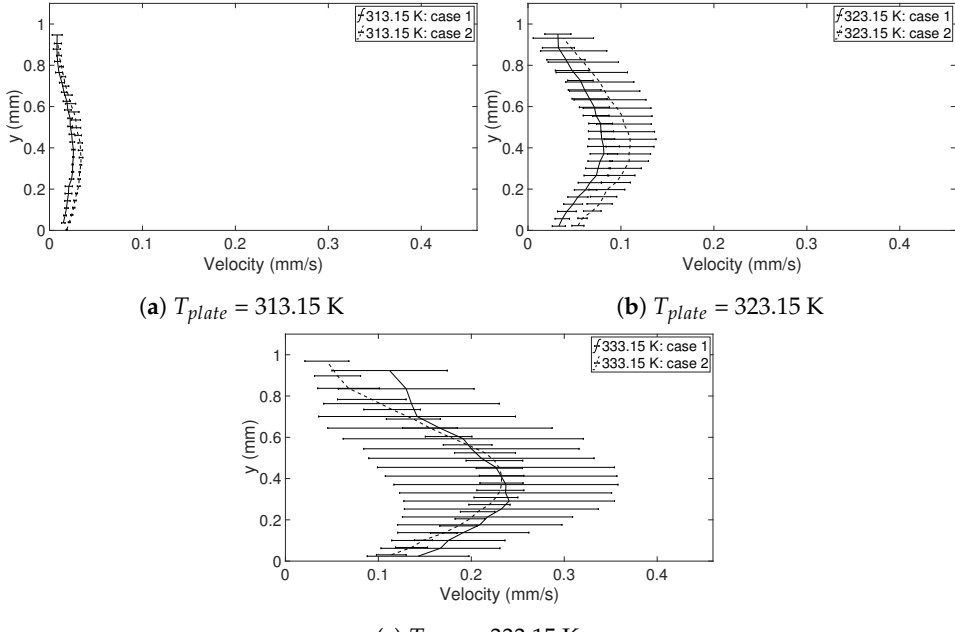

**Figure 10.** The spread (standard deviation) in velocity along the symmetry line during evaporation for $T_{plate}$ = 313.15, 323.15, and 333.15 K (all cases) when $t$ = 20–50 s shown using error bars. The solid line is the mean velocity of all times.

*3.3. Freezing*

The internal flow patterns with normalized velocity vectors for when $T_{plate}$ = 261.15 K at $t$ = 1–5 s are shown in Figure 11. Note that the velocity vectors are normalized with the magnitude of the velocity of each vector. Here, the flow is moving up along the symmetry line and down along the curved surface up to when $t$ = 0.19$t_{total}$, after which the flow changes direction and at $t$ = 0.22$t_{total}$ the flow is moving in the opposite direction—down along the symmetry line and up again along the curved surface. This flow behavior is in agreement with the flow behavior seen for $T_{plate}$ = 265.15 K by Karlsson et al. [10], but the time when the shift is slightly different. The shift occur at approximately 16% of the full freezing time when $T_{plate}$ = 265.15 K [10] compared to approximately 19% when $T_{plate}$ = 261.15 K (current study). In Figure 12, the magnitude of the velocity along the symmetry line are presented for the corresponding times in Figure 11. As for $T_{plate}$ = 265.15 K, the highest velocities are found in the beginning of freezing and the large drop in velocity that occur before the shift in the flow is also seen when $T_{plate}$ = 261.15 K. However, the increase in velocity after the shift is not as clear when $T_{plate}$ = 261.15 K as for $T_{plate}$ = 265.15 K [10], but a similar flow behavior is seen for both temperatures. The spread (standard deviation) in velocities along the symmetry line for when $T_{plate}$ = 265.15 K and 261.15 K at $t$ = 1–4 s and $t$ = 1–5 s, respectively, is found in Figure 13, where it can

be seen that the velocities in the droplet varies significantly. In the beginning of freezing, the velocities are high, but shortly after the shift in flow direction the velocities are almost zero, resulting in a very large spread in velocities with respect to time.

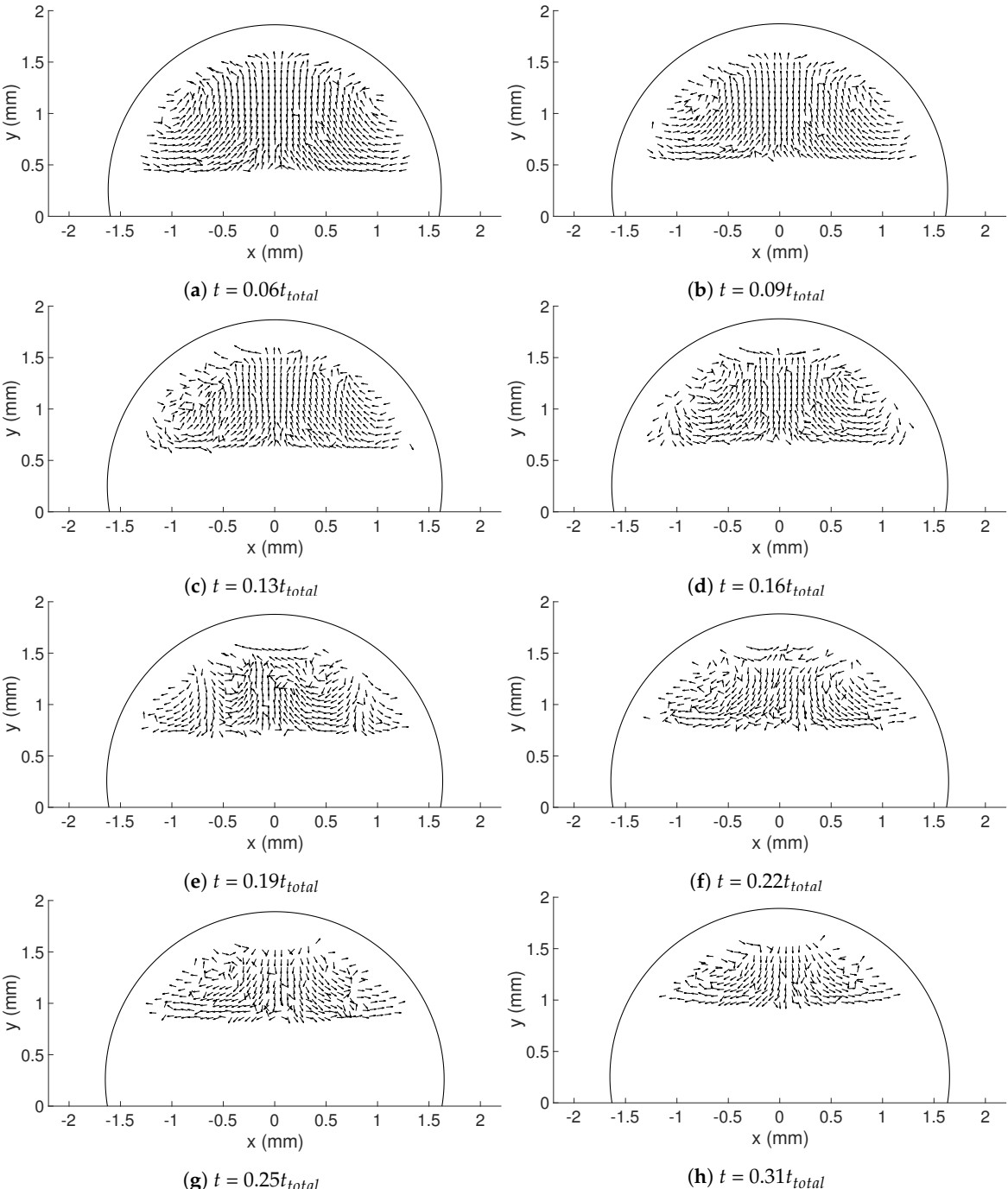

**Figure 11.** Internal flow patterns with normalized velocity vectors for the corrected data when $T_{plate}$ = 261.15 K at $t$ = 1–5 s.

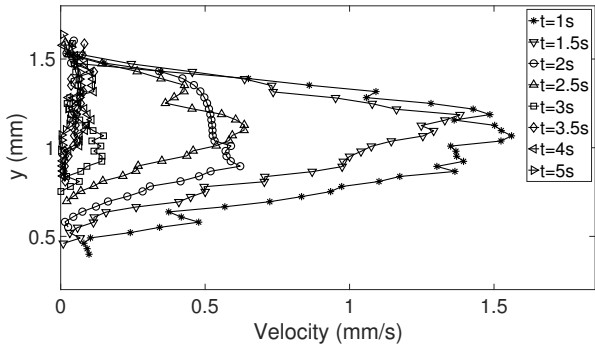

**Figure 12.** The magnitude of the velocity along the symmetry line for $T_{plate}$ = 261.15 K when $t$ = 1–5 s.

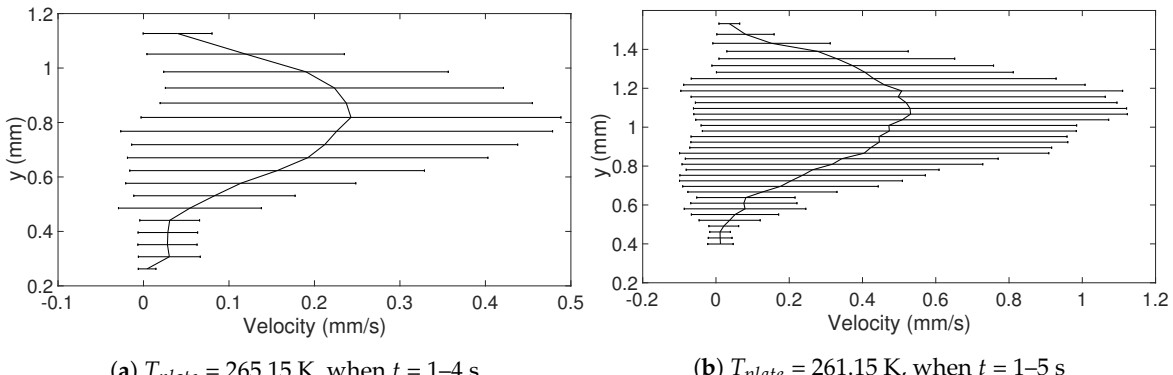

(**a**) $T_{plate}$ = 265.15 K, when $t$ = 1–4 s
(**b**) $T_{plate}$ = 261.15 K, when $t$ = 1–5 s

**Figure 13.** The spread (standard deviation) in velocity along the symmetry line during freezing for $T_{plate}$ = 265.15 K and 261.15 K when $t$ = 1–4 s and $t$ = 1–5 s, respectively, shown using error bars. The solid line is the mean velocity of all times.

### 3.4. Comparing the Flows within Freezing and Evaporating Droplets

Higher $T_{plate}$ temperatures result in higher velocities, but the highest velocities are found when $T_{plate}$ < 0. The highest velocities in this study are found when $T_{plate}$ = 261.15 K, suggesting that the velocity is increasing with decreasing $T_{plate}$ temperature for temperatures when $T_{plate}$ < 0. The maximum spread in velocities is also seen for when $T_{plate}$ = 261.15 K, suggesting that the highest velocities may be found here; however, when $t$ = $0.19t_{total}$, the velocities are almost zero, as for when $T_{plate}$ = 265.15 K [10]. Regarding evaporation, the highest velocities are found the first 15 s in all cases (see Figures 7 and 10). The highest velocities for when $T_{plate}$ = 333.15 K is comparable to the highest velocities when $T_{plate}$ = 265.15 K, suggesting that Marangoni convection might actually affect the flow, at least for the first second or seconds. For freezing droplets, Marangoni convection seems to affect the flow due to the higher velocities in comparison to evaporation. This can also be compared to the theoretical value for the Marangoni ($Ma$), which is well above 80 for all cases studied. This means that the Marangoni effects should be present in both the freezing and evaporating droplets. Finally, when the flow shifts in direction the velocity approaches zero for both evaporation and freezing and after the shift the velocities increases again, but never to the highest levels as seen before the shift in flow.

### 4. Conclusions

In this study, the internal flow in freezing and evaporating water droplets was investigated using Particle Image Velocimetry and a comparison between the two cases was performed. Three heating and two cooling temperatures were studied where the aim was to reveal if natural convection and/or if Marangoni effects influences the flow within the droplets. Due to the low viscosity in heated droplets, the early flow within these droplets might be influenced by the release of the droplet. For freezing droplets, the viscosity is high in the water and therefore the internal flow is not influenced by the release of the droplet, the movement is only due to freezing and $\Delta T$. During evaporation, within 15 s after the release of the droplet to the surface, a form of "steady-state" occurs since the conditions in and

around the droplet are not changing significantly. As for the freezing droplets, a shift in flow direction occurs early in evaporation. This might be due to the Marangoni effects, but it could not be concluded in this work. Two other explantations could be due to the temperature differences caused by the heated surface or because of the force created as the droplets impact the heated surface, pushing the water up along the curved surface. In the freezing droplets, the flow is similar between the two temperatures and shows a typical "Marangoni flow"-behavior, suggesting that the Marangoni effects influence the early flow within freezing droplets. However, the shift in direction of the flow and the magnitude of the velocities differ between the two cases. When comparing the velocities during freezing and evaporation, it was found that a warmer heating surface yields higher velocities and a larger spread in velocities compared to a colder heating temperature, although the highest velocities and the largest spread in velocities was found when the temperature of the cooling surface was below zero.

**Author Contributions:** Conceptualization, L.K., A.-L.L., and S.L.; methodology, L.K.; software, L.K.; validation, L.K. and A.-L.L.; formal analysis, L.K., A.-L.L., and S.L.; investigation, L.K.; resources, S.L.; data curation, L.K.; writing—original draft preparation, L.K.; writing—review and editing, L.K., A.-L.L., and S.L.; visualization, L.K.; supervision, A.-L.L., and S.L.; project administration, L.K.; and funding acquisition, A.-L.L. and S.L. All authors have read and agreed to the published version of the manuscript.

**Funding:** This research received no external funding.

**Conflicts of Interest:** The authors declare no conflict of interest.

## Nomenclature

| | |
|---|---|
| *A* | surface contact area ($\text{m}^2$) |
| *d* | diameter (m) |
| *h* | height (m) |
| *r* | radius (m) |
| *t* | time (s) |
| *T* | temperature (K) |
| RH | relative humidity (%) |
| *μ* | viscosity (kg/ms) |
| *Ma* | Marangoni number |
| *Stk* | Stokes number |

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
