# Peer review of "Comparing Internal Flow in Freezing and Evaporating Water Droplets Using PIV"

_water, doi:10.3390/w12051489_

Round 1

Reviewer 1 Report

The topic is interesting and it is adapt to this journal. Minor remarks:

- the abstract should be more concise, and I suggest authors provide the background, target, significance, methodology, main results, and so on, in this abstract.
- use only SI units. Now there are e.g. mm, mm2, degrees Celsius,

Reviewer 2 Report

Reviewer's comments on Manuscript ID: water-798651

Title: Comparing Internal Flow in Freezing and Evaporating Water Droplets using PIV

This research intended to investigate internal flow in freezing and evaporating water droplets experimentally by means of PIV. The paper prepared well but some issues can be addressed.

  • Since there several parameters in the text, nomenclature can be added to ease the understanding the text.
  • The experimental setup has been described adequately
  • The language of the manuscript is satisfactory
  • The results have been described sufficiently and it is conclusive.

In the current format the paper after the request changes is suitable to be publish

Reviewer 3 Report

Evaporation and freezing of droplets is two interesting areas with many applications. The internal flow within freezing and evaporating water droplets has been investigated. In general, the study is interesting, and thus I suggest accepting it after some revisions. Specific comments are given as follows.

  1. More information on the significance of investigating the internal flow within the droplets should be provided.
  2. Normalized velocity vectors are presented in Figure 5; please describe how to calculate the normalized velocity vectors. Whether it is calculated by the maximum velocity in the droplets? It is better to present a color map to show the distribution of velocity magnitude clearly.
  3. Where is the boundary of the droplets shown in Figure 5? Can you show the boundary of the droplets in Figure 5, 8 and 11?
